# The Quantification of Bacterial Cell Size: Discrepancies Arise from Varied Quantification Methods

**DOI:** 10.3390/life13061246

**Published:** 2023-05-24

**Authors:** Qian’andong Cao, Wenqi Huang, Zheng Zhang, Pan Chu, Ting Wei, Hai Zheng, Chenli Liu

**Affiliations:** 1Shenzhen Institute of Synthetic Biology, Shenzhen Institutes of Advanced Technology, Chinese Academy of Sciences, Shenzhen 518055, China; 2University of Chinese Academy of Sciences, Beijing 100049, China

**Keywords:** bacterial cell cycle, microscopic images, cell size, initiation mass

## Abstract

The robust regulation of the cell cycle is critical for the survival and proliferation of bacteria. To gain a comprehensive understanding of the mechanisms regulating the bacterial cell cycle, it is essential to accurately quantify cell-cycle-related parameters and to uncover quantitative relationships. In this paper, we demonstrate that the quantification of cell size parameters using microscopic images can be influenced by software and by the parameter settings used. Remarkably, even if the consistent use of a particular software and specific parameter settings is maintained throughout a study, the type of software and the parameter settings can significantly impact the validation of quantitative relationships, such as the constant-initiation-mass hypothesis. Given these inherent characteristics of microscopic image-based quantification methods, it is recommended that conclusions be cross-validated using independent methods, especially when the conclusions are associated with cell size parameters that were obtained under different conditions. To this end, we presented a flexible workflow for simultaneously quantifying multiple bacterial cell-cycle-related parameters using microscope-independent methods.

## 1. Introduction

Growth and division are fundamental needs of all cells. During its cell cycle, a cell needs to coordinate its growth with genome replication and cell division to achieve faithful self-replication under various conditions. In eukaryotes, the cell cycle is divided into four ordered phases, G1, S, G2, and M2. Multiple checkpoints exist to control the order and timing of cell-cycle transitions through protein phosphorylation [1]. However, in bacteria, no obvious checkpoint has been identified. During rapid growth, many bacteria can initiate new rounds of DNA replication before the completion of the previous round, resulting in overlapping cell cycles [2]. How bacteria achieve cell-cycle control to coordinate cell growth with genome replication and cell division has been the subject of frequent investigations. These investigations into bacterial cell-cycle regulation are not only helpful in controlling bacterial growth for industrial production; they may also be instructive for building a synthetic cell from the bottom up.

When considering the developmental history of bacterial physiology, the significant progress in our understanding of the bacterial cell cycle is often attributed to the improvement of relevant experimental methods and concepts [3], leading to the identification of new quantitative relations and inspiring new models. For example, by developing rigorously quantitative experimental methods and focusing on steady-state growth instead of the “obligatory life cycle” of the bacteria, Maaløe and Kjeldgaard were able to ensure high reproducibility of their experiments. Based on such reproducible quantitative data, they discovered the SMK growth law in 1958, i.e., that the population-averaged cell mass scales exponentially with the growth rate [4]. In addition, the baby machine invented by Charles E. Helmstetter [5,6] facilitated the synchronization of bacterial cell populations and enabled temporal analysis of the bacterial cell cycle [7]. By combining the baby machine with radioactive pulse labeling, Helmstetter carefully quantified the DNA synthesis rates of *E. coli* under various conditions [8,9]. These measurements provided a quantitative basis for Helmstetter and Stephen Cooper to establish the CH model, which quantitively states constant *C* and *D* periods of 40 and 20 min, respectively, for *E. coli* cells, with a doubling time of less than 60 min [2]. The *C* period refers to the period between the initiation and the corresponding termination of bacterial chromosome replication, while the *D* period refers to the interval between DNA replication termination and corresponding cell division. Subsequently, Donachie integrated the SMK growth law with the CH model and proposed the constant-initiation-mass hypothesis [10]. This hypothesis states that the replication of the chromosome is initiated when the ratio of cellular mass to the number of chromosome origins reaches a growth-rate-independent constant, termed the initiation mass (*m_i_*), and the corresponding cell division always follows DNA replication initiation by the *C* + *D* period. As this hypothesis provides a simple interpretation of how bacterial cells coordinate cell growth, DNA replication, and cell division, it has significantly impacted studies on the bacterial cell cycle. However, in repeated investigations into this hypothesis over the past few decades, both confirmation [11,12,13,14,15,16,17] and contradictions [18,19,20,21,22,23] have emerged.

While empirical observations of bulk populations have contributed to the establishment of several quantitative relationships among bacterial cell cycle parameters, the population-averaged cell behavior masks variation among individuals and does not reflect the typical behavior of single cells. Recent advances in microfluidics [24,25,26,27], high-throughput imaging [28,29], and automated image analysis [30,31,32] have enlivened the study of single-cell bacterial physiology [33] and provided novel opportunities to explore problems that are challenging at the population level, such as cell-size homeostasis. Through the dynamic tracking of numerous cells with single-cell resolution, the universal strategy for bacterial cell-size maintenance known as the “division adder correlation” has been discovered [34,35,36]. Furthermore, the combination of single-molecule fluorescent labeling and single-cell tracking has significantly facilitated the investigation of chromosome organization [37,38,39,40], replisome dynamics [22,41,42,43], and stochasticity or noise in the bacterial cell cycle [44,45,46]. By fluorescently labeling the relevant molecules of different cell cycle events, the cell cycle progression in individual bacterial cells can be monitored. These long-term observations can generate a large amount of single-cell quantitative data that aid in identifying correlations between different cell-cycle events and in uncovering quantitative laws of cell-cycle control [14,17,47,48,49]. As an example, in 2019, Si et al. employed the fluorescently labeled replisome protein to visualize replication cycles and investigated both the division adder and initiation adder under various perturbations [47]. More recently, Govers et al. quantified and analyzed broad phenotypes of the fluorescently labeled *E. coli* and numerous gene deletion derivatives in various media using microscopic images, then identified four new quantitative relations that were related to nucleoid segregation and different steps of cell division [17].

As part of the advancements in single-cell related techniques, many types of software tools have been developed to facilitate the high-throughput and automated extraction of cell-cycle-related parameters of bacterial cells from microscopic images. These software tools have been widely adopted in many studies. However, little attention has been paid to the impact of using different software, or the same software with different parameter settings, on the results and on the relevant conclusions. This paper demonstrates that discrepancies exist when analyzing identical datasets with different software or with the same software with different parameter settings. Importantly, these discrepancies can lead to different conclusions when validating quantitative relations, even if the consistent use of a particular software and specific parameter settings is maintained throughout a study. Therefore, it is recommended that conclusions be cross-validated using microscope-independent methods, and a flexible workflow is presented for this purpose.

## 2. Quantification Methods Based on Microscopic Images

Due to the presence of the diffraction limit and the small size of bacterial cells, accurately determining the actual boundary of bacterial cells in microscopic images can be challenging. A variety of high-throughput software has been developed to automatically obtain the properties of bacterial cells [32]. Generally, the pipeline for automatic cell-size evaluation includes image brightness correction, cell segmentation, and morphology extraction [30]. The cell segmentation is pivotal for high-quality cell-size characterization. Current bacterial cell segmentation algorithms broadly fall into two categories: classical computer vision and machine-learning-based algorithms. The former requires manual optimization of tunable parameters through the visual inspection of segmentation results, as was carried out by MicrobeJ [50], Oufti [51], BacStalk [52], CellProfiler [53], and CellShape [54]. Machine-learning based algorithms rely on training with labeled ground-true data and their performance is largely dependent on the quality and size of the training dataset. Among such algorithms, deep neural networks (DNNs) have emerged as superior tools for cell segmentation [55,56]. As several excellent studies have comprehensively introduced or compared these algorithms/software tools [31,32,56,57,58,59], we will not conduct a quantitative evaluation of their segmentation quality here. Instead, we focus on discussing the influence of software and parameter settings on quantitative outcomes when the segmentation results are satisfactory.

To demonstrate this, the following experimental and analytical procedures were implemented. First, for reliable quantification of cell size, it was necessary to establish a steady-state growth status of the cells. Otherwise, significant variations in the results of characterizing cell-cycle-related parameters may have occurred when samples were taken at different time points [60,61,62]. Therefore, we captured phase-contrast images of *E. coli* K12 substr. NCM3722 grown in four different media. In brief, the steady-state growth was established by serial dilution, as previously described [23]. The cells were immobilized using a 1% agarose pad (prepared with 0.9% NaCl (*w*/*v*)) when OD_600_ reached ~0.2. The immobilized cells were imaged within 5 min at room temperature (RT), using an inverted microscope (IX-83, Olympus, Tokyo, Japan) equipped with a 100× oil objective (Olympus), an automated xy-stage (ASI, MS2000), and a sCMOS camera (Prime BSI, photometrics). Three types of software, MicrobeJ, Oufti, and BacStalk, were selected to process these images. Various parameter settings were achieved by adjusting the *auto-threshold offset* of MicrobeJ and the *cellwidth* and *meshwidth* of Oufti. For BacStalk, we used its default setting. The satisfactory segmentation performance was verified through visual inspection (Figure 1a) and the outlier were excluded by manual correction or filtered according to intensity and cell area. Cell size parameters, including cell length, cell width, and cell area, were obtained directly from the software output. In addition, we developed customized image-processing scripts based on deep-learning algorithms. The processing pipeline can be summarized in four steps: first, segmenting individual cells using U-Net [63,64]; second, determining edge details using Otsu’s thresholding; third, calculating the midlines of cells through interpolation; and last, calculating size parameters including length, width, and area. Except for Oufti, the cell volume (*V*) was calculated by the software based on cell length (*L*) and width (*W*) and the formula $V=\frac{4}{3}\pi {\left(\frac{W}{2}\right)}^{3}+\pi {\left(\frac{W}{2}\right)}^{2}\left(L-W\right)$, assuming that *E. coli* is a cylinder with hemispherical polar caps. All of these size parameters, which represent the population-averaged values for more than 4500 cells in each growth condition, are listed in Table 1.

It is apparent that the absolute values of cell-size parameters, such as cell length, cell width, area, and volume, are affected by the software and the parameter settings. Therefore, investigators should use consistent criteria, including the same software and parameter settings, to process microscopic images in a study. However, we questioned whether this alone was sufficient to produce conclusive results.

To this end, we considered the validation of the constant-initiation-mass hypothesis as an example here. This hypothesis proposes that the initiation mass, which refers to the cellular mass per *oriC* at the time of replication initiation, remains constant at different growth rates. To validate this hypothesis, investigators should assess the initiation mass of wild-type cells cultivated in diverse growth media exhibiting varying growth rates. The assessment of initiation mass can be carried out by employing time-lapse images of cells that have been fluorescently labeled to indicate replication initiation events, or by utilizing snapshot images in combination with techniques that facilitate the quantification of the population-averaged *oriC* number. Here, we employed the latter method, since we already had the cell volume data in four different growth media. The population-averaged *oriC* numbers were quantified by analyzing the DAPI-stained samples of run-out experiments with flow cytometry, as described previously [23].

The conclusion regarding the validation of the constant-initiation-mass hypothesis is affected by the software and parameter settings utilized for the analysis of the microscopic images. We calculated the initiation mass ($m_{i}$) based on the widely used equation [65], $m_{i}=\frac{\bar{V}}{\bar{o}}\times \frac{1}{ln2}$, where $\bar{V}$ and $\bar{o}$ are the population-averaged cell volume and the *oriC* number, respectively. As shown in Table 1, the absolute value of the cell volume, i.e., $\bar{V}$, is largely affected by the software and parameter settings. Thus, the absolute value of $m_{i}$ is also subject to these effects. More importantly, as the relative cell volume across different growth conditions is also influenced by the software and parameter settings, the relative initiation mass across the four media exhibited different trends depending on the software and parameters employed. Upon calculating the relative initiation mass based on cell volume data obtained via MicrobeJ, with an auto-threshold offset set to −200, we found a gradual increase in relative initiation mass as growth rates increased from 0.1 to 0.9 h^−1^, with cells grown in MOPS + glucose exhibiting a notable ~50% increase, compared to cells in MOPS + glutamine. This implied that the initiation mass was not constant, but growth-rate-dependent (Figure 1b, left panel). However, divergent conclusions may emerge when adopting an *auto-threshold offset* setting of 100. In that case, the calculated relative initiation mass varied by only ~10% between cells grown in MOPS + glucose and MOPS + glutamine (Figure 1b, second panel). When using Oufti for image processing, adjustments to cell width and mesh width parameters yielded similar discrepancies. The relative initiation mass displayed a growth-rate-dependent or growth-rate-independent pattern, depending on whether parameter Set1 or Set2 was used, respectively, for image processing (Figure 1b, third and fourth panel). Since the Set1 and Set2 in Oufti can produce almost the same cell length, the observed change in trends was mainly attributed to the variation in cell width. These findings suggest that the constancy of initiation mass can be influenced by software and parameter settings. Given that the time-lapse imaging approaches for quantifying the initiation mass also need to calculate the cell volume based on microscopic images, the effect of software and parameter settings are expected to be same.

Collectively, these results strongly imply that when drawing conclusions that rely heavily on comparing the sizes of cells cultured under different growth conditions, using the same software and consistent parameter settings for image analysis is not sufficient to produce conclusive results, and additional efforts are required to enhance the credibility of the conclusion. One possible solution is to calibrate the parameter settings of the specific software using standard nanoparticles with a known diameter that is comparable to the cell width. However, it is noteworthy that variations in the optical properties of bacterial cells and nanoparticles can still give rise to inconsistencies. Therefore, we recommend cross-validation of the conclusions, whenever possible, using techniques that are not dependent on microscopic images.

## 3. Quantification Methods Not Reliant on Microscopic Images

This section provides an overview of microscope-independent techniques that are capable of measuring the cell size, the cellular *oriC* number, and the initiation mass. By integrating these methods, we introduced a flexible workflow for concurrently quantifying these parameters (Figure 2). This workflow can be utilized either individually or for the corroboration of findings obtained from microscopic images.

To establish a steady-state growth, bacteria should be maintained in exponential growth for at least 10 generations via serial dilution (Figure 2a), and the growth rates of the total biomass and the cell numbers should be monitored to verify the steady-state growth (Figure 2b). Once the steady-state growth is established and verified, samples for quantifying cell-cycle-related parameters can be taken at any time point, as the average cell composition per cell is expected to be constant [66].

The population-averaged cellular mass ($\bar{m}$), which is closely related to the cell volume, can be characterized by dividing the total biomass by the total cell numbers of the population. Dry weight and OD are two common metrics for quantifying bacterial biomass. Compared with the dry weight measurement, determining the OD of liquid cultures with a spectrophotometer was more convenient (Figure 2c). Plate counting and flow cytometry (Figure 2d) are two methods for the absolute enumeration of bacterial cells. The plate counting often requires serial dilution of the cell culture to ensure a countable range of colony numbers (25–250 colony forming units, or CFU, bacteria on a standard petri dish) [67]. Compared with plate counts, flow cytometry requires sophisticated hardware, but it is a faster and more accurate technique for measuring cell densities. For an apparatus with a controllable sample flow rate, e.g., CytoFLEX (Beckman Coulter), the cell concentration of samples can be conveniently measured with appropriate dilution and staining methods [23]. If the sample flow rate of the flow cytometer is unknown, suspensions of microspheres with standard densities must be used as references [68]. It is worth noting that, although the forward scatter (FSC) determined by flow cytometry can also reflect the relative size of the cell, it is difficult to compare the FSC obtained from different instruments, even when using the same parameter settings [23].

The population-averaged cellular *oriC* number ($\bar{o}$) can be determined through a run-out experiment and, by combining this with $\bar{m}$, the initiation mass ($m_{i}$) can also be obtained. To carry out a run-out experiment, cephalexin and rifamycin were added to the cell suspension to inhibit cell division and DNA replication initiation, respectively. Cells were then allowed to grow under the same culture conditions for 2–3 mass doubling time to complete the on-going replication [23]. Consequently, the number of fully replicated chromosomes after run-out was equal to the number of *oriC* at the time of the addition of the compounds (Figure 2e). After fixation with 70% ethanol and staining with appropriate DNA dye (such as DAPI), the samples of the run-out experiments could be analyzed with flow cytometry or a fluorescence microscope. It should be noted that, rifamycin-resistant replication initiation will invalidate the run-out experiment in several genetic backgrounds [69,70], and the mechanism for this resistance is still unclear. According to derivation process presented in [65], once we have $\bar{o}$ and $\bar{m}$ for cells in a steady-state growth status, $m_{i}$ can be calculated using the following equation: $m_{i}=\frac{\bar{m}}{\bar{o}}\times \frac{1}{ln2}$.

Consider the aforementioned validation of the constant-initiation-mass hypothesis as an example. Both $\bar{m}$ and $\bar{o}$ are found to be positively correlated with the growth rates. Specifically, when compared with cells grown in MOPS + glutamine, the $\bar{m}$ and $\bar{o}$ of cells grown in MOPS + glucose increased 2-fold and 1-fold, respectively (Figure 3a,b). As a result, rather than remaining constant, the initiation mass ($m_{i}$) was growth-rate-dependent. It increased continuously as the growth rates increased, exhibiting a ~50% increase in cells grown in MOPS + glucose compared to those cultivated in MOPS + glutamine (Figure 3c), which was comparable to the result obtained based on microscopic images using MicrobeJ with the auto-threshold offset set to −200. Therefore, it can be concluded that the initiation mass of *E. coli* K12 substr. NCM3722 cells is not constant, but dependent on the growth rate when grown within the range of 0.1 to 0.9 h^−1^.

Two points should be noted at the end of this section. First, as the optimal approach may vary depending on the individual characteristics of each case, in this study, we did not intend to endorse the utilization of any specific software or parameter setting for accurately quantifying cell volume based on microscopic images. Second, the presented microscope-independent workflow was derived from our daily practice and may be limited, to some extent, by our scope of knowledge or the available instruments. Other well-established methods that have not been mentioned in the current workflow can certainly be adopted for cross-validation purposes or used directly to address certain scientific questions.

## 4. Conclusions

Microbiology is primarily an experimental science, and the use of different experimental methods may sometimes generate conflicting conclusions for the same scientific question. Therefore, researchers should understand the pros and cons of the various methods to effectively design experiments and to evaluate results. In this paper, we found that the application of different software and different parameters for image analysis can yield variations in both the absolute and relative sizes of cells grown in various conditions, leading to divergent conclusions. Therefore, it is of paramount importance to employ microscope-independent approaches to cross-validate the conclusions drawn solely from image analysis, especially when it comes to a quantitative comparison of cell volume across various perturbations.

## Figures and Tables

**Figure 1 life-13-01246-f001:**
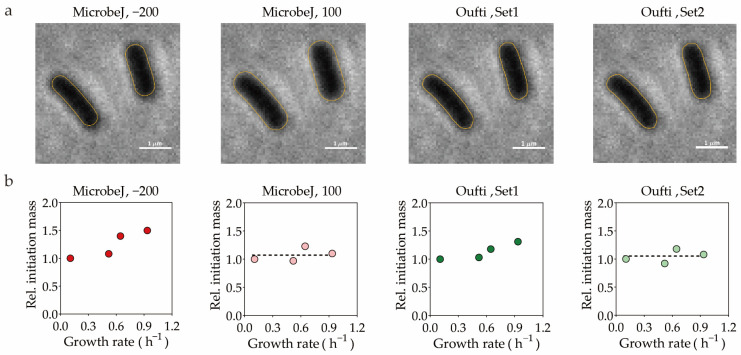
Different software and different parameter settings can yield divergent conclusions. (**a**) The detected contours of cells grown in MOPS medium with glutamine as the sole carbon with different parameters and software are shown by yellow lines. With MicrobeJ, we took *auto-threshold offset* = −200 and *auto-threshold offset* = 100 as an example. For Oufti Set1, we set *cellwidth* of cells grown in MOPS + glutamine, MOPS + alanine, MOPS + glycerol, and MOPS + glucose at 8, 9, 9, and 12, and for Oufti set2, we set *cellwidth* at 9, 9, 10, 11. The values of *meshwidth* were set to the corresponding *cellwidth* plus 2. The additional parameters are documented in Appendix A Appendix A and were maintained consistently for both Set1 and Set2. (**b**) The relative initiation mass obtained by different software and different parameter settings. The relative initiation mass was calculated based on cell volume data obtained with different software and different parameter settings, and the population-averaged *oriC* number (see Section 3). The horizontal dashed line represents the average relative initiation mass for cells in four growth conditions.

**Figure 2 life-13-01246-f002:**
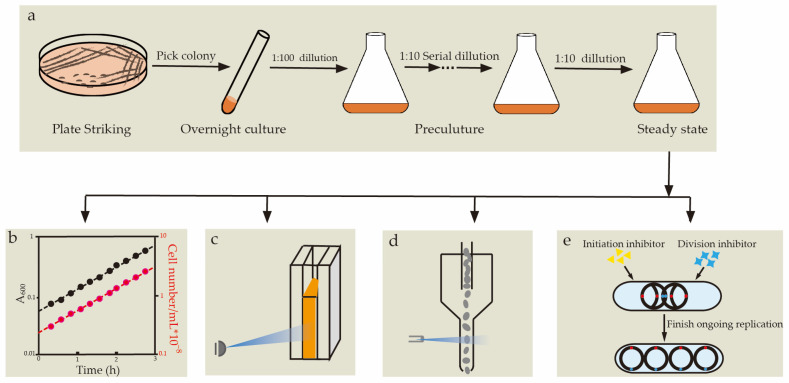
A workflow for simultaneous quantification of multiple parameters at the population level. (**a**) Establishment of the steady-state growth by serial dilution. (**b**) Verification of the steady-state growth by monitoring the growth rate of total biomass and cell number. The cell number (red lines) and cell mass (black lines) growth curves can form two parallel lines in semi-log plots if the steady-state growth has been achieved. (**c**) OD measurement. (**d**) Cell counting by flow cytometry. (**e**) Quantification of averaged cellular *oriC* number by runout experiments.

**Figure 3 life-13-01246-f003:**
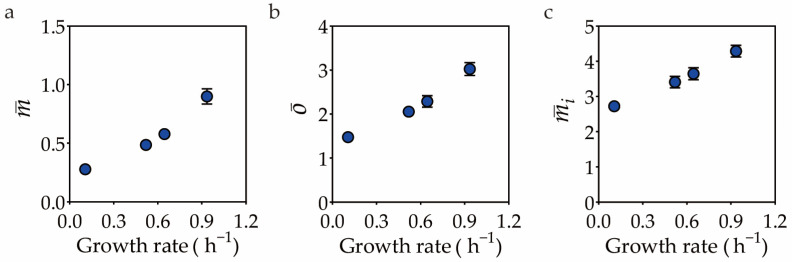
Cross-validation of the growth-rate-dependence of the initiation mass by the microscope-independent method. Growth rate dependence of the population-averaged cellular mass (**a**), population averaged cellular *oriC* number (**b**), and initiation mass (**c**). The $\bar{m}$ and $\bar{o}$ are characterized as described previously [23]. The units for $\bar{m}$ and $m_{i}$ are OD_600_ mL per 10^9^ cells and 10^−10^ OD_600_ mL, respectively. Error bar represents the standard error of biological replicates.

**Table 1 life-13-01246-t001:** Cell features quantified with different software or parameters.

Features ^1^	Medium ^2^	MicrobeJ	MicrobeJ	Oufti	Oufti	BacStalk	Custom
		−200	100	Set1	Set2		Scripts ^3^
Length (µm)	Glutamine	1.85 ± 0.44	2.16 ± 0.47	2.03 ± 0.47	2.04 ± 0.47	2.25 ± 0.48	2.11 ± 0.54
	Alanine	2.22 ± 0.59	2.51 ± 0.67	2.38 ± 0.56	2.38 ± 0.56	2.63 ± 0.66	2.35 ± 0.45
	Glycerol	2.61 ± 0.67	2.94 ± 0.72	2.77 ± 0.65	2.78 ± 0.64	3.04 ± 0.69	2.79 ± 0.88
	Glucose	2.73 ± 0.65	2.97 ± 0.68	2.81 ± 0.67	2.80 ± 0.66	3.11 ± 0.67	2.88 ± 0.60
Width ^4^ (µm)	Glutamine	0.55 ± 0.06	0.75 ± 0.06	0.52 ± 0.03	0.55 ± 0.03	0.79 ± 0.06	0.55 ± 0.05
	Alanine	0.61 ± 0.06	0.81 ± 0.06	0.57 ± 0.02	0.57 ± 0.02	0.83 ± 0.06	0.60 ± 0.05
	Glycerol	0.67 ± 0.08	0.88 ± 0.08	0.60 ± 0.03	0.64 ± 0.03	0.90 ± 0.07	0.65 ± 0.05
	Glucose	0.79 ± 0.08	0.96 ± 0.08	0.72 ± 0.04	0.69 ± 0.03	1.00 ± 0.07	0.74 ± 0.06
Area (µm^2^)	Glutamine	0.94 ± 0.26	1.51 ± 0.37	1.04 ± 0.25	1.10 ± 0.27	1.17 ± 0.26	1.02 ± 0.22
	Alanine	1.28 ± 0.36	1.90 ± 0.54	1.35 ± 0.33	1.35 ± 0.33	1.47 ± 0.36	1.25 ± 0.24
	Glycerol	1.66 ± 0.48	2.46 ± 0.66	1.64 ± 0.42	1.74 ± 0.43	1.86 ± 0.44	1.60 ± 0.30
	Glucose	2.03 ± 0.52	2.70 ± 0.67	2.00 ± 0.51	1.92 ± 0.49	2.13 ± 0.47	1.90 ± 0.39
Volume ^5^ (µm^3^)	Glutamine	0.39 ± 0.13	0.85 ± 0.24	0.44 ± 0.11	0.49 ± 0.13	0.96 ± 0.25	0.46 ± 0.13
	Alanine	0.59 ± 0.19	1.15 ± 0.37	0.63 ± 0.16	0.63 ± 0.16	1.26 ± 0.36	0.59 ± 0.12
	Glycerol	0.85 ± 0.31	1.62 ± 0.51	0.80 ± 0.22	0.90 ± 0.23	1.74 ± 0.48	0.84 ± 0.26
	Glucose	1.21 ± 0.37	1.92 ± 0.56	1.18 ± 0.31	1.09 ± 0.29	2.18 ± 0.57	1.11 ± 0.25

^1^ The length, width, area, volume presented in the table correspond to the average value of these parameters of the cells in the population. ^2^ Four media refers to MOPS media with glucose, glycerol, alanine, or glutamine as the sole carbon source, with corresponding growth rates of 0.93  ±  0.06, 0.65  ±  0.02, 0.52 ±  0.01, 0.11  ± 0.02 h^−1^, respectively. ^3^ The custom scripts have been made openly available and can be accessed through the link provided in Appendix A Appendix A. ^4^ MicrobeJ provides the mean width of the cell as the cell width. BacStalk employs the maximum width of the cell body and our custom scripts used the fitted mean cell width. In Oufti, we defined cell width by the mean width of cell mesh, as Oufti does not directly provide cell width. ^5^ For Oufti, we used the cell volume directly provided by the software instead of that calculated by the length and width.

## Data Availability

Data associated with this manuscript are available in the Appendix A.

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
