# Peer review of "The Quantification of Bacterial Cell Size: Discrepancies Arise from Varied Quantification Methods"

_life, 2023, doi:10.3390/life13061246_

Round 1

Reviewer 1 Report

General remark

In this interesting opinion-article, the authors have analyzed the influence of 4 software packages on cell size measurements like length, width, and area, obtained from microscopic images. An outcome of their study is that the initiation mass as a function of growth rate can either increase or remain constant, depending on the software applied to the same cell images.

To validate the different software tools, the authors recommend the use of microscope-independent methods. Illustrated by a workflow (Figure 2), important advises are given for culturing and measuring cell populations, like verification of the status of steady-state growth. Such practice (as performed in ref. 23), is most lacking in numerous recent studies.

Specific comments and questions

Line 136 and Table 1: "Cell size parameters including cell length,...." How were the cells (in Figure 1a) prepared for microscopy (agarose pads?) and photographed (phase-contrast?). Does "Length (µm)" apply to the average length in the population? How many cells were counted?

Line 144: "...assuming E. coli is perfectly rod-shaped." Cell shape in the formula applies to a cylinder with hemispherical polar caps.

Line 167, Table 1, last column: What signifies "Custom scripts"?

Line 203: The absolute values for cell volume (Table 1) and oriC number (Figure 3) are calculated. How do these values compare to those obtained in other studies (eg. refs. 12 and 21)?

Line 246: "However, in practice, it is recommended to take the sample at a relatively high optical density (OD), such as OD600 ~ 0.2, at which the steady-state growth status is not affected."

Although in ref. 23 it was shown that this practice works, I think this recommendation is undesirable. It has been general practice to use OD 460nm! See, for instance, ref. 19 (Bremer-group). Because a shorter wavelength gives a higher dispersion of the light and thus a higher OD-value, lower cell concentrations can still be acurately measured with the spectrophotometer. This is important for obtaining steady-state growth because cell populations at low concentrations are far from becoming depleted of nuctrients and thus from reaching stationary phase.

Editorial comments

- Line 19: "The precise regulation of the cell cycle..."  Or is it rather "The robust regulation of the cell cycle..."  ?

- Line 29 and throughout: "microscope-independent methods".

- Line 120: "excellent literatures" > "excellent studies" ?

Reviewer 2 Report

In this Opinion manuscript Cao and colleagues report findings of cell length measurements using some commonly used program packages such as Oufti and MicrobeJ, and the effect of different parameters on cell length measurements. Especially parameters such as cell volume can differ significantly by relatively small changes in cell length and width, potentially affecting analysis and conclusions derived from these measurements. The main point raised is that if conclusions are derived from the automated analysis of microscopy images, it will be wise to employ an additional approach not based on microscopy to cross-validate the conclusions drawn. The work then also describes a number of different techniques that might be suitable.

Overall the manuscript is quite far developed. The main conclusion in some respect is an obvious one that good-quality research always should rely on: key findings should be verified by using at least two different experimental techniques. Very often this approach not only cross-confirms conclusions, but also enriches the conclusions, as the data sets from different approaches more often than not are richer and for this reason allow more detailed conclusions than just a single approach. Thus, the main conclusion is valid and there is no harm of pointing this out.

My main criticism is the approach taken by the authors. For example, the threshold setting in MicrobeJ simply defines on which side of the high contrast edge the cell boundary is defined, with negative values resulting in a cell shape inside of the high contrast edge, and positive values resulting in cells being defined as slightly outside of the high contrast edge. This is also clearly shown in the images provided where the outline is shown for a -200 and +100 setting. It is pretty self-explanatory that, based on these settings, cell length and width measurements will differ. So what? As long as these threshold settings are consistently used, they simply will result in a change from the “real” cell length/width by a constant factor. Similarly, different algorithms will determine cell shapes slightly differently, so there is no surprise that different programs and bioinformatics approaches will result in different measurements. The main conclusion here should be presented more clearly, which is that researcher need to be careful to not vary algorithm and specific settings when trying to compare data sets. This would be a much more important point than the notion that with different settings the precise measurements indeed vary, which is pretty much given.

One additional point the authors are trying to raise is that additional derived measurements, such as cell volume, will be more strongly influenced by what might seem like small changes to measurements such as cell length and width, as volume is obviously cubed, resulting in bigger changes. The authors point out that different settings can result in data sets that would be interpreted differently if looked at in isolation. While possible, parameters such as threshold should result in a fixed factor change, and the observation that a different setting/software can result in data that would be interpreted differently seems rather specialised. In fact, if changes as small as reported by the authors (even though I do not doubt that they are real!) can change the interpretation of the results, then there is a far bigger problems, as the experimental approach obviously is simply not reliable enough to allow any conclusions being drawn. Or, to use a quote by famous chemist Ernest Rutherford: “If your experiment needs statistics, you ought to have done a better experiment”.

Where does this leave us with the manuscript? I appreciate that the authors have done a very good job in developing the Introduction, with a wide variety of relevant papers and a historic overview that is very nice to read and insightful to younger researchers. This alone deserves to be published. Also, as already suggested, the main conclusions are valid, even if they are pretty obvious to more senior researchers. There still is no harm in pointing them out to students and less experienced colleagues. However, in my opinion the “data development” section can be quite drastically shortened. Examples can be given, rather than the systematic presentation of data, as all the data do not add anything to the main conclusion. Also, the authors absolutely have to comment on what some of the parameters are. For Oufti they simply use “set 1” and “set 2”. What do these settings do? Are they similar to the relatively straight-forward threshold setting that MicrobeJ is using? In which case this should be explained. Or are they representing more complex calculations? If so again this needs to be explained. The more of a “Blackbox” such parameters or parameter sets are, the more care must be taken to not introduce accidental or deliberate variation, as otherwise data sets cannot be compared. I do appreciate that the authors have taken a practical approach based on their specific work, for which only a limited understanding of what precisely these parameters are doing is needed, but the current descriptions remain pretty meaningless to the reader. I think the Introduction, with a much shortened “data” section that highlights approaches and potential pitfalls much more directly than in the current version, together with a good summary of what other approaches there are for cross-validation, will change this into a useful short Opinion paper.

Minor comments

All references ought to have spaces between the last word of the sentence and opening bracket. In its current format the lack of a space is making the text hard to read, and unnecessarily so.

Line 40, “overlapping cell cycles” will need a reference

Line 121, colloquialisms such as “won’t” should be avoided and written in full.

Line 177, “It is obvious that, the” – remove the comma.

Line 203, even if the calculation is wide used references still should be given.

The manuscript reads well, with only few and minor grammar problems and typos.

Reviewer 3 Report

The study by Cao et all takes a systematic approach to investigate the effects of software and parameter settings when quantifying microscopy data.  This is a critical factor to consider when designing and interpreting data, and ensuring robustness when microscopy is used in combination with other techniques.  

Overall, the paper is well presented. The major issues for me are a lack of information on methods including:

how were the images captured (microscope and objective specifications)?

how were steady state experiments set up? What OD did you sample at?

Details on population-averaged oriC experiments are lacking - referenced to section 3 but not clear if same protocol as ref 23 is used, and if DAPI staining was employed.  If so, how was this imaged and quantified?

Are the custom scripts publicly available? Settings for this, as well as BacStalk, are missing from supplementary information.

Dataset S2 was not available for me to download.

Minor comments:

It is worth mentioning in the introduction/discussion that cell cycle progression and quantification has focused on fast growing rod shaped bacteria, and that the data presented in this study may not apply to slow growing or coccoid bacteria.

Line 134: states that segmentation was visually confirmed. How did you control for user bias? Was this a blinded comparison of software?

Table 1: Width is defined by different measurements e.g. mean or maximum width. How does this affect comparison between software?

Figure 1: what does horizontal line represent on graphs?

Reference 2 is incomplete

Overall the article is well written and is clear and concise.  

Minor comments:

Line 161: should be in MOPS, not on MOPS

Line 180: Using the constant... this is a redundant sentence that doesn't make sense by itself, and should be incorporated into the following sentence. 

Line 288: Specifically.  Should this be a comma rather than a full stop? 

Line 294: Comparable with that obtained... no capital letter? Or is this a new sentence? 
